# Impact of the COVID-19 pandemic on young people from black and mixed ethnic groups' mental health in West London: a qualitative study

Romane Lenoir  , Keri Ka-Yee Wong

Psychology & Human Development, University College London, London, UK

**Correspondence to**
Dr Keri Ka-Yee Wong;
keri.wong@ucl.ac.uk

## ABSTRACT

**Objectives** The COVID-19 pandemic has disproportionately impacted vulnerable groups' physical and mental health, especially young people and minority ethnic groups, yet little is known about the crux of their experiences and what support they would like. To address this gap, this qualitative study aims to uncover the effect of the COVID-19 outbreak on young people with ethnic minority backgrounds' mental health, how this changed since the end of lockdown and what support they need to cope with these issues.

**Design** The study utilised semi-structured interviews to conduct a phenomenological analysis.

**Setting** Community centre in West London, England.

**Participants** Ten 15 min in-person semistructured interviews were conducted with young people aged 12–17 years old from black and mixed ethnic groups who regularly attend the community centre.

**Results** Through Interpretative Phenomenal Analysis, results indicated that the participants' mental health was negatively impacted by the COVID-19 pandemic, with feelings of loneliness being the most common experience. However, positive effects were concurrently observed including improved well-being and better coping strategies post lockdown, which is a testament to the young people's resilience. That said, it is clear that young people from minority ethnic backgrounds lacked support during the COVID-19 pandemic and would now need psychological, practical and relational assistance to cope with these challenges.

**Conclusions** While future studies would benefit from a larger ethnically diverse sample, this is a start. Study findings have the potential to inform future government policies around mental health support and access for young people from ethnic minority groups, notably prioritising support for grassroots initiatives during times of crisis.

## INTRODUCTION

The SARS-CoV-2 (COVID-19), infecting over 600 million people and claiming more than six million lives and counting, has triggered unprecedented economic and social disruptions worldwide.[1 2] While everyone has been impacted, certain groups have been disproportionately affected,[3] including individuals

---

**STRENGTHS AND LIMITATIONS OF THIS STUDY**

⇒ This qualitative interview study during COVID-19 gives voice to the experiences of young people from black and mixed ethnic backgrounds in the UK.

⇒ The in-person quality of the interviews helped build rapport between the researcher and the young people and sharing of sensitive issues around mental health access and support, increasing the results' validity.

⇒ This is a convenient sample, with girls and those aged 15 years and above being disproportionately represented in our data, as they provided most of the answers.

⇒ The small sample size and lack of ethnic diversity limits the generalisability of the study to individuals from other ethnic minority groups.

---

from ethnic minority groups.[4] Mortality rates in the UK between March and April 2020 for instance were two and three times higher for individuals from minority ethnic groups than for ethnically white groups.[5] In April 2021, the UK Intensive Care National Audit and Research Centre reported that ethnic minorities represented 34% of the patients in intensive care units, despite only constituting 11% of the UK population.[6] These alarming trends of physical vulnerability for individuals from ethnic minorities during COVID-19 begs the question of why this is the case, and what support might look like for this group.

An ecological explanation can be offered to elucidate these patterns. Indeed, a holistic consideration of the dynamic interplay between an individual and their environment[7 8] suggests that situational risk factors such as multigenerational living conditions and environmental adversity—most notably associated with individuals from minority ethnic groups' living circumstances—may increase ethnic minority groups' vulnerability to COVID-19.[9] Additionally, they are over-represented in socially disadvantaged

neighbourhoods, with white individuals only representing 8.7% of the population in the 10% of the most income-deprived neighbourhoods in the UK in 2016.[10] These are characterised by a limited capacity to socially distance, resulting in higher COVID-19 infection and death rates.[11] Ethnic minority individuals also occupy more key worker positions, as they constituted over 20% of the National Health Service workforce while representing under 15% of the working-age population in the UK in 2019.[12] This overall disadvantage in the labour market exposes them to greater risk during the COVID-19 pandemic.[13] Finally, it may also be the case that social networks being racially differentiated increase intracommunity transmission for ethnic minority groups.[14] The COVID-19 pandemic thus not only exacerbated pre-existing health inequalities for ethnic minority groups in the UK, but also deepened socioeconomic ones, the impact of which must be investigated.[15]

Studies of the effect of the COVID-19 pandemic on ethnic minority groups' day-to-day life stressors have been particularly informative. Mind, a UK-based mental health charity, surveyed adults' experiences during the pandemic and found that individuals from minority ethnic groups experienced higher levels of everyday distress.[16] Overall, 30% of adults from ethnic minorities reported that their mental health was affected by housing issues, compared with 23% of whites. Similar rates were reported for troubles with employment (61% against 51% for white individuals), finances (52% against 45%) and physical health support (39% vs 29%).[16] However, ethnic minorities notably represented less than 5% of the study's sample. Significantly, within individuals from ethnic minority groups, the pandemic seemingly impacted the mental health of individuals from black and mixed ethnicities the most.[17] Additionally, individuals from ethnic minorities in the UK were more exposed to food insecurity during COVID-19 and had a higher reliance on food banks than their white counterparts.[18] Finally, long-term inequalities in socioeconomic opportunities and a history of mistrust towards governing entities have cultivated feelings of alienation among minority ethnic groups.[19 20] The COVID-19 pandemic deepened this rift, reinforcing feelings of social isolation.[14 21]

Young people from ethnic minority groups have also suffered during COVID-19, being deprived of structures essential to their socioemotional development, which caused increased psychological distress.[22–24] Yet, young people have also demonstrated considerable resilience, most importantly through protective family environments and increased self-awareness.[25 26] Documenting these experiences is crucial, as young peoples' voices are too often missing from academic research on COVID-19 and mental health due to difficulties in recruitment and access. Their scarcity makes developing support mechanisms for young people rather difficult. Also, as a majority of studies are quantitative in nature[27 28] which, although informative, does not create a space for young people to voice their needs and allow them to feel heard. Not least,

young people from minority ethnic groups are at heightened risk of experiencing distress during the COVID-19 pandemic and therefore deserving to be actively involved in and listened to in this research.

Existing quantitative studies on the mental health of individuals from minority ethnic backgrounds are limited in that they very rarely examine the experiences underlying the patterns they observe. To date, very few qualitative studies have focused on the impact of COVID-19 on minority ethnic individuals, and even less on young people. The first is Mahmood *et al*'s paper,[29] which interviewed 19 ethnic minority community leaders between October and November 2020 about the COVID-19 pandemic's impact on their communities. Participants spoke about the financial, social and physical consequences of the pandemic, the latter being subjectively justified by historical and structural disparities. Results indicated that mental health was a fundamental concern, as the participants invoked their group's usual tight-knit nature to illustrate the extent of lockdown's impact on their well-being.[29] Second, Burgess *et al* investigated ethnicity and mental health during COVID-19, with a highly relevant target demographic of young people, with 87% of black participants.[30] Through four focus groups, the study identified deteriorating mental health and experiences of racism, alongside more encouraging patterns of resilience and growth through self-care. Other qualitative studies on young people's mental health during the pandemic, although not specifically focusing on ethnic minorities, similarly note worrying patterns of worsening well-being and specifically increased anxiety, racism and stigma.[31 32] The scarcity of the literature solely concentrating on first-hand experiences of minority ethnic groups throughout the COVID-19 pandemic highlights the need for more qualitative studies to truly understand how this pandemic has impacted this group.

With the pinnacle of the pandemic now behind us, we must identify what the barriers to mental health support might be and how we can best support young people now rather than later. First, recognising that young people's struggles are legitimate and encouraging them to seek support is essential.[16] Second, the structural and interpersonal obstacles young people from ethnic minority groups face when seeking mental health support are a key factor driving low incidences of access to support.[33] Within the Mind survey, half of the young people revealed that difficulties in accessing support made their mental health worse.[16] Lower English proficiency has also been associated with reduced use of mental health resources, with a significantly steeper effect within minority ethnic groups.[34] The low intelligibility of online information surrounding COVID-19 in the UK revealed a striking lack of appropriate education materials available to these groups.[35] Intracommunity stigma, most significant for black individuals in the UK, is also a long-standing impeding change in mental well-being.[36] Further isolating young people, these obstacles exacerbate their distrust, and create a reluctance and stigma around

seeking mental health support.[29] The reality is that young people's mental health as a whole, but most significantly those from ethnic minority groups, has not only been worsening since the start of the COVID-19 pandemic but is also poorly assisted.[29 37 38]

The current qualitative interview study aims to identify the impact of the COVID-19 pandemic on young people from minority ethnic groups and the types of support they might need. By giving voice to communities less heard from, we hope to inspire larger-scale research and develop tailored mental health support for black and mixed ethnic minority communities.

## METHODS

### Participants

Ten participants were recruited via convenience sampling from a community centre in West London. The current preregistered CopeWell Study[39] (https://osf.io/jcak7/), funded by UK Research and Innovation, aimed to understand the impact of the COVID-19 pandemic on young people's mental and physical health and to cocreate appropriate life-skill workshops to enable young people from black and mixed ethnic backgrounds to better cope with the pandemic. The present study focuses on the initial interviews on the impact and challenges of COVID-19 on young people's lives. A pilot study was conducted for the interview schedule, confirming that the questions' content and order were cohesive (online supplemental appendix A). The prods' usefulness in allowing the participants to elaborate was assessed, familiarising the researcher with their navigation. Parents received and completed an information sheet and consent form before the study started. Similarly, participants' written consent was obtained at the time of the interview. They were also reminded of their right to withdraw or omit to answer at any point. Confidentiality was ensured and potential questions were answered. The study was then conducted from 17 October 2021 to 8 December 2021, where participants took part in 15 min one-on-one semistructured interviews (M=14:06 min) with 14 open-ended questions. They were thanked and debriefed post interview and received a £30 gift voucher as an honorarium. We built rapport with the youth workers and young people before conducting the interviews at the community centre (a familiar and safe environment). COVID-19 safety measures (eg, masking, distancing, well-ventilated space) were applied where necessary. The participants' answers were recorded via audio software to maximise accuracy and were subsequently transcribed by RL.[40]

### Materials

The initial pilot interview schedule contained 14 open-ended questions, with prods guiding the discussion according to the participants' reactivity and answers, and was designed by KK-YW. It was subsequently amended in consultation with RL to form the final version (online supplemental appendix B). Questions progressed from items about the COVID-19 pandemic and its impact on the participants' mental health, to potential changes in these experiences. This was followed by the kind of support they received and needed to cope with these struggles.

### Data analysis

The analysis was conducted following Smith's Interpretative Phenomenological Analysis (IPA).[41] IPA is relevant for this study as it draws focus on the young people's points of view and the meanings they ascribe to their situation. The explicitly idiographic nature of this analytical method allowed for the participants' lived experiences to be thoroughly probed into.[42] Smith and Shinebourne's six steps analysis for IPA was followed,[43] resulting in the delineation of three superordinate themes, each containing themes and subthemes. Through an iterative process of reading and re-reading the transcripts, the researcher (RL) made initial annotations and colour-coded quotes according to the study aims they pertained to. Coding was continued until saturation was reached (70% of transcripts).

The most frequent and significant patterns were then clustered together (online supplemental appendix C) and agreed with KK-YW through discussions. The thematic relationship was summarised in an initial table with subthemes and sample quotes (online supplemental appendix D). IPA, as an iterative process, allowed the researcher to continuously revise themes and quotes, concomitantly ensuring the integrity of the participants' accounts was preserved.[43] Applying this not only preserved the interpretative role of the researcher in IPA, but also buffered researcher bias risks.[44 45] The final thematic relationship is presented in a table (online supplemental appendix E).

### Patient and public involvement

Patients or the public were not involved in the design, or conduct, or reporting or dissemination plans of our research.

## RESULTS

A total of 10 young people aged 12–17 participated in the study (M=14.3, SD=1.3 years, girls=70%). Participants self-identified from a list provided in the questionnaire with the following ethnic groups: black or black African (n=4), black or black Caribbean (n=1), any other black background (n=1), white and black African (n=2) and white and black Caribbean (n=1). One participant preferred not to say. Three superordinate themes emerged, namely overall declining patterns of mental health during the COVID-19 pandemic, improvements post lockdown and needs moving forward.

**Table 1** Impact of the COVID-19 pandemic on mental health

| Theme | Subtheme | Illustrative quotes |
|---|---|---|
| Individual impact | Self-esteem | P103: 'There was a lot of challenges with like my mental health and like my self-esteem and like I- I still don't but like, I never believed in myself like at all, like at all/ (…) I just like feel like I was never going to be good enough'. (Age 15)<br>P104: 'I realised that I had a lot of mental health problems within myself, and I wasn't not really happy, um not very confident, quite insecure.' (Age 16) |
| | Lack of motivation | P107: 'I feel like because I haven't been doing a lot I've lost the motivation to actually do things'. (Age 15)<br>P107: 'Um mentally, I think yeah as I said before my motivation has taken like a big decline and also obviously I feel like bad habits during COVID there's not really much that you can do'. (Age 15)<br>P110: 'A lot, uh yeah like laziness, um can't get out of bed as much.' (Age 15) |
| Symptoms of mental illness | Anxiety, depression and disordered eating | P100: 'Mentally, it affected me both like oh am I going to get COVID next, like got some sort of anxiety.' (Age 15)<br>P103: 'I lost a lot of weight after the pandemic not in the best way. I didn't really eat (…) I never really used to give myself like the attention I deserve and you know with COVID it made it very very bad like my mental health was very bad, (…) and it was so such a heavy weight on my heart, it was like it was really hard for me and COVID made it all worse (…) so I was losing it, I was losing it.' (Age 15)<br>P104: 'I realised a lot of things which kind of drove me into a… I would say quite a depression in the sense of like I would just staying in my room the whole time, I wouldn't leave it.' (Age 16) |
| | Trauma | P104: 'Nobody was grasping with the fact that people are really dying, (…) like it was sad that so many people couldn't handle COVID they took their lives like it was…it was absolutely devastating.' (Age 16) |
| Relational | Isolation and individualisation | P103: '(Lockdown had a) big impact on my mental health like it made me feel way more alone that like I actually was cause I couldn't go outside obviously. (…) I also felt isolated and also felt like alone and unable to be myself in my own space.' (Age 15)<br>P104: 'Cause you can have like a million people around you and still feel alone. So I felt like in that moment I felt so alone and…tired and mentally exhausted (…) But I feel like everybody was more out for themselves and not really trying to really help each other (…)… I feel like the community was kind of broken.' (Age 16) |
| | Positive solitude | P100: '(I enjoyed) having me time. Like being able to do like stuff on my own.' (Age 15)<br>P107: 'Um, things I enjoyed the most I'd probably say I got a lot of time to myself.' (Age 15)<br>P104: 'I used to go for like walks around my neighbourhood (…) like I just used to reflect and it was such a like a peaceful time like after I came back home I thought like…I just felt like free. I felt like there was no **** nothing.' (Age 16) |
| | New connections | P101: 'I met lots of new people and got to interact with some people during lockdown.' (Age 12)<br>P104: '(People could) talk to each other, sit down and really think like what is going on in the world.' (Age 16)<br>P104: 'I definitely lost friendships and gained friendships.Like I found true friends and I also lost people that I thought were good friends to me.' (Age 16) |

## Superordinate theme 1: impact of the COVID-19 pandemic on mental health
### Individual
As a result of being required to stay indoors, 30% of the young people described suffering from decreased self-esteem during the COVID-19 pandemic. Underlying insecurities manifested themselves through self-doubt and low confidence. Additionally, 20% of the young people spoke about the monotony of life during lockdown causing them to lose their motivation and ambition (table 1).

### Symptoms of mental illness
Participants' previous apprehensions about life and patterns of low mood were intensified by the COVID-19 pandemic, as 60% of them recounted symptoms of anxiety, depression and disordered eating. Lockdown freed up time for overthinking and the development of unhealthy habits of thoughts and behaviour. These were heightened by anxieties surrounding the illness itself for 20% of the participants, through exposure to a number of traumatic events such as suicides and climbing death rates (table 1).

### Relational
The isolation provoked by lockdown shifted the participants' rapport to others, negatively impacting 90% of the young people's mental health. Loneliness and fears arising from the speed and risk of virus transmission fuelled tensions and individualisation trends among ethnic minority groups. However, solitude was also perceived as restorative, as 30% of the young people used this time to reflect and enjoy time on their own as a break from an otherwise hectic life. Interestingly, 50% of the participants also reported that the COVID-19 pandemic not only allowed them to reinforce current connections and friendships but to develop new ones (table 1).

## Superordinate theme 2: changes in mental health since the end of lockdown
### Autonomous adaptive coping skills
Being left alone with their thoughts led the young people to demonstrate great levels of resilience in the face of adversity. Indeed, 30% of them reported introspective self-evaluations and the development of adaptive self-regulating mechanisms. Overall, 50% of the participants observed growth in themselves, transpiring into

**Table 2** Changes in mental health since the end of lockdown

| Theme | Subtheme | Illustrative quotes |
|---|---|---|
| Autonomous adaptive coping skills | Overthinking | P102: 'I had to do it because if I didn't then how am I supposed to go in life so I literally just looked at the present and just focusing on what I need to do to get through my life.' (Age 15)<br>P103: 'Overthinking like I started to like really, evaluate my personality and like who I am as a person.' (Age 15) |
| | Better coping | P103: 'Those little things that I did like I used to go on walks or I just used to like I used to write down my thoughts a lot like it was just like random bits of what I'm feeling throughout the day and like those kind of help you feel better.' (Age 15)<br>P103: 'When I realised it's like there so many different things that I can do to help myself and I just want to find all the right um outlets to project any um anger or aggression any uh sad feelings that I had without causing violence.' (Age 15) |
| | Personal growth | P102: 'I've been depressed I know I've been all down and whatever, but I think if I didn't have that stage in my life I would not be what who I am right now.' (Age 15)<br>P103: 'And um like I think it strengthened my relationships but it also strengthened the relationship that I had with myself which is like one of the most important one.' (Age 15)<br>P109: 'Back then I wasn't as social (…) Cause I found a really good series of books and I just kept reading it.(…) It's not that I couldn't be social it's just that I wasn't bothered to, but now I'm like talking a bit more, still reading though.' (Age 14) |
| Lifted restrictions | Freedom | P103: 'At the end of lockdown when everything like um the restrictions started lifting I felt like I was finally getting life back.' (Age 15)<br>P106: 'Going out in groups without having to get COVID tested themselves before they left the house.' (Age 13)<br>P107: 'I took things for granted because obviously something as simple as being outside and going to the shops it's all changed.' (Age 15) |
| | Social bonds | P104: 'Seeing my friends for the first time. Like I can't fake that that was like the best feeling cause it's like all these months I haven't seen you and it's like oh my god.' (Age 16)<br>P107: 'when the restrictions first eased like in the summer of 2020. (…) Yeah I feel like then I got a lot closer with a lot of people.' (Age 15) |

their relationships with others. This demonstrates great resilience and resourcefulness among the young people (table 2).

### Lifted restrictions

Lockdown ending meant newfound freedom, which 50% of the participants experienced as positively impacting their mental health. Restrictions being lifted allowed them to enjoy normal life again by being outside and regaining individual freedoms. Additionally, social bonds were reinstated, further benefitting their mental health (table 2).

### Superordinate theme 3: limited support obtained during the COVID-19 pandemic and current needs
#### Support

Overall, 70% of the young people deplored a lack of support during the COVID-19 pandemic. In terms of structural support, 50% of the participants were critical of the academic help they were provided with. Additionally, 30% of them shared feeling neglected by society and governing bodies, evidenced by a deep mistrust in national institutions and strong feelings about being overlooked, ignored and misunderstood as young people from black and mixed ethnic minority backgrounds (table 3).

#### Needs

The young people requested various forms of support moving forward, the largest being psychological (50%). In response to social bonds being challenged during the COVID-19 pandemic, 40% of the participants also requested relational support. With socialising being key to their well-being, support in this area would benefit them greatly. Finally, practical support with professional opportunities and maintaining a healthy routine, habits which were abruptly perturbed by the COVID-19 pandemic, was considered useful by 50% of the young people (table 3).

### DISCUSSION

This study investigated the impact of the COVID-19 pandemic on the mental health of young people from black and mixed ethnic groups, how this has changed since the end of lockdown and the support that they may need to cope with these issues. Results indicated that young people's mental health was both negatively and positively impacted by the COVID-19 pandemic. However, their mental well-being seemingly improved after the peak of the COVID-19 pandemic, due to adaptive coping mechanisms being developed and lockdown ending. Finally, the young people deplored a lack of support during the COVID-19 outbreak and were critical of the structural support they did receive. They asked for

**Table 3** Limited support obtained during the COVID-19 pandemic and current needs

| Theme | Subtheme | Quotes and illustrations |
|---|---|---|
| Support | Lack of support | P100: 'Honestly little to none, support yeah.' (Age 15)<br>P110: 'I wouldn't say there was support, it was just really independent.' (Age 15) |
| | Neglect | P102: 'I would actually listen to people because clearly the government doesn't listen to no one he just he does everything what he thinks and I don't I don't trust that.' (Age 15)<br>P102: 'I know I have issues (…) It's just like, I need more support with that. (…) No one is listening. So I have to shout it out loud so everyone can listen.' (Age 15)<br>P104: 'In this generation and age there are so many misconceptions about teenagers, especially me as a black female teenager, there's many misconceptions about…being…you know who you are or what you look like.' (Age 16) |
| | Structural | P100: 'in school, it's very much like,…once in a blue moon ask us if we're alright and that's it.' (Age 15)<br>P106: 'They don't help they only sit in my class and do nothing when I ask for help then sometimes they listen.' (Age 13) |
| Needs | Psychological | P104: 'Sometimes you just, you just need an ear. You just need someone to listen. I feel like that would really help actually. And it would've helped I think back then if I'd had that I think it would have been way easier to talk about.' (Age 16)<br>P104: 'And (the therapist) just said what you described does sound like (depression), and I was like, in that moment like a tear fell on my face.' (Age 16)<br>P106: '(I would need) Like people speaking to me.' (Age 13) |
| | Relational | P105: 'I would have like hugs from my friends. I feel like it's a way of comforting myself and comforting other people when it's like hardships and stuff, it just like helps a lot because even the simplest thing, like love can do a lot.' (Age 13) |
| | Practical | P107: 'I would say maybe more opportunities available. So stuff like sports. Even something like work experience. Sort of that yeah I feel like that would help a lot cause it would help like ease back into things (…) making new friendships, relationships, has become a lot harder.' (Age 15)<br>P100: 'About how to maintain a healthy lifestyle even though…circumstances can change like this (snaps fingers).' (Age 15) |

psychological, relational and practical support moving forward.

First, participants reported experiencing the negative consequences of the COVID-19 pandemic on a personal level, in their relationships and on their mental health. This reinforces past research evidencing deteriorating mental health in young people from minority ethnic groups, both quantitatively[24 27 46] and, to a lesser extent, qualitatively.[25 30] More specifically, such patterns have been observed in the ethnic minority groups prevalent in our study. Mixed ethnic young people seemed to have most challenges with mental health and social connections, while black young people struggled with academics.[47] The present study nonetheless also demonstrated an improvement in the participants' mental health post pandemic (November–December 2021), echoing wider studies reflecting this.[25 26 29] McKinlay et al.'s results were not ethnic specific,[25] increasing the present study's relevance in focusing on the experiences of young people from black and mixed ethnic minority groups. Additionally, the measures used in Penner et al's research were taken before the studied area's COVID-19 peak,[26] limiting their validity. The present study, however, interviewed young people who have experienced multiple lockdowns, strengthening the results' validity. However, reports of past experiences are subject to recall bias and individuals may overestimate positive affect retrospectively as a coping strategy, which potentially reduces the internal validity of the present study's results.[48 49]

The improvements in mental health post lockdown found in our study demonstrate great resilience among the young people. This is supported by research evidencing how young black men in particular exhibit resilience when faced with hardship by drawing on different sources of social capital.[50] However, our results contrast with other literature suggesting that, although observing a general amelioration in mental health symptoms after lockdown ended, scores remained higher than pre pandemic.[51] This implies a long-lasting detrimental impact, which was not observed in the present group of young people. This is potentially due to measurement differences as most quantitative studies measure changes in scores while qualitative ones focus on the most salient experience for the individual. Additionally, research suggests that although improvements in mental health after lockdown were observed among most young people, they did not manifest in at-risk youth.[52] The latter study's selection criterion for at-risk youth was household income, which although partially overlapping with ethnic minority, remains a distinctive non-interchangeable demographic.[53] This is not a criterion we used for our young people.

Finally, the young people deplored a lack of support during the COVID-19 outbreak and were critical of the structural support they received, echoing previous studies

deploring poorly adapted mental health resources for individuals from ethnic minorities.[16 35] Indeed, these highlight issues of stigma and the ways in which non-culturally sensitive forms of therapy and imbalanced power dynamics affect the therapeutic relationship, as reported among black participants.[54] This creates further reluctance to seek help for this particular ethnic group. Additionally, participants expressed feeling neglected by society and governing entities, mirroring previous reports of alienation among individuals from minority ethnic groups.[14 21] The present study's qualitative methodology thus effectively contributed to solidifying the existing body of literature around this theme. Study participants did not mention English proficiency as a barrier to them accessing mental health support, as was suggested by Sentell *et al*'s research.[34] However, the participants are native English speakers, suggesting this obstacle may not apply to individuals from all ethnic minority groups. Similarly, stigma around mental health was not explicitly identified as an obstacle to mental health access, contrary to Mahmood *et al*'s findings.[29] In fact, study participants advocated for mental health awareness and its importance for public health discussion. One reason for this discrepancy in findings could be because participants in Mahmood *et al*'s study were adult community leaders,[29] a significantly older generation compared to the young people in our study, potentially reflecting a generational difference in perceptions of mental health, stigma and willingness to discuss health issues. Fortunately, this might indicate that young people from black and mixed ethnic groups simply need more exposure to mental health knowledge overall and opportunities for open, non-judgmental discussions.

## Strengths and limitations

A key strength this study possesses is its qualitative nature and in-person data collection method. By allowing young people to elaborate on their personal experiences, this study captured subtleties that wide-ranging mental health scales may have overlooked.[55] Additionally, as COVID-19 studies were mostly run mid-pandemic, sanitary restrictions compelled them to be conducted remotely.[30] Although online and anonymised methods for interviewing have been associated with lower social desirability, they increase the risk of measurement error and misclassification.[27 56] Virtual environments curb the effectiveness of non-verbal cues, affecting rapport which is vital when conducting sensitive research with vulnerable populations.[57 58] The present study's in-person collection method of qualitative data enabled the researcher to invest time and effort in building rapport with the participants, increasing the results' validity and providing the participants with a valuable space to feel heard.

However, the wide range and variation in the study participants' age represents a key limitation. Indeed, participants ranged from 12 to 17 years. The convenience sampling method used gave the researcher no control over the specific ages of the young people included in the study. This is relevant as young people's experiences from late childhood to late adolescence vary widely.[59] During this period, they go through a time of significant personal development, within which age is a highly influential factor.[22] Their age, and associated experiences, thereby inevitably introduce bias in the stories they told and the needs they voiced. That said, we recognise that our results do not reflect answers from the full age sample, containing mostly quotes from participants aged 15 and above, and encourage future research to further this endeavour with a larger representative sample. Additionally, we note that we do not elucidate on gendered patterns, as we did not predict that our sample would be imbalanced. However, we acknowledge that there is a large body of existing evidence investigating this particular factor and its interaction with ethnicity and mental health across the pandemic.[17 47 60] The results' generalisability to young people from minority ethnic groups overall is therefore low, nonetheless capturing relevant perspectives from young people of black and mixed ethnic minority groups.

## Implications for practice and research

Young people's needs—psychological, relational and practical—encapsulated in this study resonate with the current initiatives and policies being put into place by organisations, but should be sustained post pandemic. Initiatives such as telehealth for young people from ethnic minority backgrounds[61] and the youth COVID-19 Support Fund[62] allowed for the continued provision of support and safe spaces for young people during the COVID-19 pandemic. However, as black young people have reportedly struggled most with remote communications among ethnic minority groups, there is a striking need for targeted support.[47] Unfortunately, schemes supporting ethnic minority groups were highly hindered by lockdown and social distancing, limiting their scope for action.[63] Importantly, young people's involvement in the creation of frameworks that will impact them was identified by the United Nations Convention on the Rights of the Child as a fundamental human right.[64] Young people's voices and needs for support identified in the present study already had a wide ranging although short-term impact across stakeholders involved. Basing the CopeWell Study[39] workshops on young people's voices allowed previously unnoticed needs to be addressed, such as relational support, mental health and practical help to enter the professional world. The charity—who engages with the young people on a weekly basis—gained a deeper understanding of the young peoples' needs and expanded the scope of possible initiatives beyond psychological support.[16] Student researchers involved also gained first-hand experience with a full timeline of the research process and insight into the widely differing experiences of individuals throughout the COVID-19 pandemic, which adds to the existing literature.[3] By bringing previously unnoticed voices into focus, in cooperation with the CopeWell project,[39] this study exemplifies how community-based research can be a means to advocate for greater policy developments promoting racial justice in response to

the COVID-19 outbreak.[65] Crucially, this study highlights the importance of developing grassroots support that is sustained through policy and practice for young people from black and mixed ethnic groups in the aftermath of the COVID-19 pandemic and beyond.

Future research should proactively identify those unanswered needs and provide support, particularly for black and mixed ethnic young people through local coproduced solutions. Specifically, prior research solely focuses on assessing attitudes towards available support or lack thereof, which may be less adaptable to future changes and needs. Conversely, our qualitative interviews provided the young people with a platform to share the assistance they would want. Through the process of giving voice to young people, they reported feeling heard and seen,[66] giving strength to qualitative methods of inquiry considering long-term perspectives. However, this study is not without limitation. Participants' ethnicities were limited to individuals from black and mixed ethnic backgrounds and a wide age range, offering an incomplete picture of young people from ethnic minority groups' mental health in the COVID-19 context.[67] This highlights the need for future research to include varied samples of young people from minority ethnic groups in their qualitative investigations, yielding results with higher validity. Alternatively, age groups may be considered individually to tailor results and subsequent interventions to potential corresponding variations in needs. Studies may also extend to wider geographical areas, investigating cultural variations in the impact of the COVID-19 pandemic on young people from minority ethnic backgrounds, raising awareness towards this hitherto neglected issue on a larger scale. Finally, this study evidenced that the COVID-19 outbreak was cruelly reflective of social disparities in the UK, warranting the need to probe into other vulnerable groups' mental health in this context, namely individuals with disabilities or lower socioeconomic statuses.

## CONCLUSION

This qualitative interview study of young people from black and mixed ethnic backgrounds during the COVID-19 pandemic has highlighted important challenges and barriers to mental health access and young people's need for support with mental health during COVID-19, as assessed in October–December 2021. Unexpectedly, study findings also highlighted young people's resilience and lessons learnt when faced with global world challenges, such as coping mechanisms and the establishment of new relational connections. The study yielded an in-depth qualitative understanding of black and mixed minority ethnic young people's needs and support together with input from charity workers, student coresearchers and academics. Future research studies with larger, more representative samples can enhance our understanding of the hardships vulnerable groups face in such unprecedented times, the strength and resilience they have and

can inform the development of support specific to the unjust burden they carry.

**Acknowledgements** RL would like to extend sincere thanks to her supervisor KK-YW for her advice and continuous support, as well as to the young people and social workers, Yara Mirdad, CEO of Jamal Edwards Delve, who took part in this study for sharing their time, experiences, and stories.

**Collaborators** Jamal Edwards Delve Charity London (Yara Mirdad, CEO)

**Contributors** RL was involved with the data collection, analysing and interpretation of data, drafting, writing, reviewing and final approval of the article. KW was involved with the conception of the research question, data collection, drafting, writing, reviewing, management of the project, supervision of the work and final approval of the article. RL and KW agree to be joint gaurantors of the work conducted and accountable for all aspects of the work in ensuring that questions related to the accuracy or integrity of any part of the work are appropriately investigated and resolved.

**Funding** This work was supported by UK Research and Innovation, from UCL's HEIF Knowledge Exchange and Innovation Fund 2021-2022.

**Competing interests** None declared.

**Patient and public involvement** Patients and/or the public were not involved in the design, or conduct, or reporting, or dissemination plans of this research.

**Patient consent for publication** Not applicable.

**Ethics approval** This study involves human participants and was approved by UCL Institute of Education Ethics Committee (REC 1558) on 29 October 2021. Participants gave informed consent to participate in the study before taking part.

**Provenance and peer review** Not commissioned; externally peer reviewed.

**Data availability statement** Data are available in a public, open access repository. Data are available in a public, open access repository, OSF. https://osf.io/jcak7/.

**ORCID iDs**
Romane Lenoir http://orcid.org/0000-0001-6277-8203
Keri Ka-Yee Wong http://orcid.org/0000-0002-2962-8438

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
