## [Reviewer comments · BMJ Open]

ARTICLE DETAILS

TITLE (PROVISIONAL)	The impact of the Covid-19 pandemic on young people from black and mixed-ethnic groups' mental health in West London: A qualitative study
AUTHORS	Lenoir, Romane; Wong, Keri Ka-Yee

VERSION 1 – REVIEW

REVIEWER	Webber, Ruth Bradford Institute for Health Research
REVIEW RETURNED	28-Feb-2023

GENERAL COMMENTS	The impact of the Covid-19 pandemic on young people from black and mixed-ethnic groups' mental health: A qualitative study. REVIEW This piece is competently written and makes a strong case for the need to hear about experiences of the impact of the pandemic on young people from minority ethnic backgrounds which are often excluded from the literature. The paper provides good context setting for the specific impact on minority ethnic groups. You are reflective about the strengths and limitations, and provide interesting discussion about implications for policy and practice. However, I feel that there are changes which need to be made in order for this to be publishable. My primary concerns relate to referencing. You write in paragraph 2 on page 8: 'Existing quantitative studies on the mental health of individuals from minority ethnic backgrounds are limited in that they very rarely examine the underlying nature of the relationships. To date, only one qualitative study has focused on the impact of Covid-19 on minority ethnic individuals to elucidate 'why' certain relationships exist.' Firstly, what relationships are you referring to? This is a confusing paragraph and loses the thread of what your paper is trying to do. Also just from a quick google scholar search I have found a range of studies that have in fact looked at this issue from a qualitative perspective (see below for just some examples). For a paper to be of a publishable standard it needs to demonstrate rigour of research of previous work in the field in order to highlight its contribution. As it stands, claiming only one qualitative study has looked at the link between ethnicity and the impact of covid-19 on mental health is untrue and suggests a lack of extensive reading. o Shah, P., Hardy, J., Birken, M., Foye, U., Rowan Olive, R., Nyikavaranda, P., ... & NIHR Mental Health Policy Research Unit
--

Covid coproduction research group. (2022). What has changed in the experiences of people with mental health problems during the COVID-19 pandemic: a coproduced, qualitative interview study. *Social Psychiatry and Psychiatric Epidemiology*, 57(6), 1291-1303.

o Gillard, S., Dare, C., Hardy, J., Nyikavaranda, P., Rowan Olive, R., Shah, P., ... & NIHR Mental Health Policy Research Unit Covid coproduction research group Katie Anderson Nick Barber Anjie Chhapia Beverley Chipp Tamar Jeynes Jo Lomani Karen Machin Kati Turner. (2021). Experiences of living with mental health problems during the COVID-19 pandemic in the UK: a coproduced, participatory qualitative interview study. *Social psychiatry and psychiatric epidemiology*, 56, 1447-1457.

o <https://www.mind.org.uk/news-campaigns/news/existing-inequalities-have-made-mental-health-of-bame-groups-worse-during-pandemic-says-mind/>

o Pearcey, S., Burgess, L., Shum, A., Sajid, E., Sargent, M., Klampe, M. L., ... & Waite, P. (2023). How the COVID-19 pandemic affected young people's mental health and wellbeing in the UK: A qualitative study. *Journal of Adolescent Research*, 07435584231151902.

o Scott, S. R., Rivera, K. M., Rushing, E., Manczak, E. M., Rozek, C. S., & Doom, J. R. (2021). "I hate this": A qualitative analysis of adolescents' self-reported challenges during the COVID-19 pandemic. *Journal of Adolescent Health*, 68(2), 262-269.

Furthermore, on P9 you say: 'The reality is that ethnic minority young people's mental health has not only been worsening since the start of the Covid-19 pandemic but is also poorly assisted [36, 37].' But if you claim only one study has looked at this, the claim you make here would not have enough evidence. Furthermore, neither reference 36 or 37 make any reference to minority ethnic young people's experiences. Again you talk about 'studies of the effect of the Covid-19 pandemic on ethnic minority groups' day-to-day life stressors', citing a qualitative piece of work by Mind (which I would explain who they are when you introduce them as knowledge that they are a UK-based mental health charity is currently assumed). Finally, in the discussion you say that your findings: 'reinforces past research evidencing deteriorating mental health in young people from minority ethnic groups, both quantitatively [25, 28, 44] and qualitatively [26]', the very research you claim has not been done apart from one qualitative study. This is further evidence for the fact that your introductory discussion on existing work needs significant changes to be made, to acknowledge that there is qualitative work in this area, which as it stands it does not, yet goes on to seemingly reference work that has done this.

The sample size is very small (10) and the length of the interviews (20 minutes) renders the data collected very minimal. While qualitative research is by its nature small-scale and although you have clearly drawn patterns across the data, this seems particularly minimal, generating only 200 minutes worth of data. Again I can only speak from my background of anthropology, sociology and social policy, but this is a very limited amount of data to be able to draw conclusions from.

The structure of the paper, in particular the table's layout and use of themes and sub-themes almost identically mimics the qualitative paper that is the main focus by Mahmood et al. 2021. I am not from a health sciences background so it may be that this is

	standard practice, however you use the same quotes in 'connections' part of table 1 and 2 but where 1 is talking about during the lockdown, 2 is talking about after the lockdown, so these quotes can't be used to say the same thing as they are talking about two separate time periods. You discuss different explanations for disproportionate impact of pandemic on minority ethnic populations, writing that: 'a biological explanation would suggest that ethnic minority groups have a higher risk of developing stress-related physiological responses and comorbidities [7, 8]. The explanations which follow this are much more robust in terms of addressing structural issues, while this first 'explanation' is very deterministic. Again I am not from a medical background but I'm not sure it has been claimed that minority ethnic groups are inherently more predisposed to poor mental health. I would delete this sentence and the following discussion is stronger, as this sentence suggests. You use the term 'co-production' in the conclusion but this is a very specific approach not used in this research and I would remove reference to this as this is the first and only mention of this and the research is not co-produced. Again, this paper is for the most part very well-written, but these issues above need to be addressed in order for it to be a publishable standard, and to ensure the continuity of a high standard of rigorous research.
--	---

REVIEWER	Bamford, Jordan The University of Manchester Division of Psychology and Mental Health
REVIEW RETURNED	02-Mar-2023

GENERAL COMMENTS	Dear Editor, thank you for the opportunity to peer review this paper. The paper I have reviewed presents as a qualitative study which aims to explore the impact of the covid pandemic on ethnic minority youth mental health, and identify what support may best serve this population. This study included 10 participants recruited via convenience sampling in West London, a majority of the sample were black. Participants took part in semi-structured interviews. Interviews were analysed via an interpretative phenomenological framework. This study identified interesting insights into the impact of covid on mental health, in particular the individual impact on self-esteem, motivation, experience of anxiety, depression and eating disorder, exposure to trauma and isolation. Further this study identified changes in mental health since the end of lockdown, in particular aspects of personal growth and better coping is reported by the youth in this study. This study explored the support given during the pandemic, and despairingly shows that young people felt there was a lack of support and felt neglected. This study shows evidence for the negative impact of covid on young people's mental wellbeing, but also speaks to the resilience and coping strategies of young ethnic minority groups. This is an innovative paper as it explores the particular views of young black minorities which are often excluded in academic literature. Overall I think this paper is valuable and offers important insights. My academic background relates to determinants of ethnic minority mental health, so I have focussed my review on
--

this area, and less on the qualitative methodology, although I identified no significant concerns. I have a few suggestions relating to the background and discussion which I believe would strengthen this paper:

Major areas:

1. I noted in the methodology and results section there was no mention of saturation of themes with respect to this qualitative study. Can authors please indicate if saturation of themes was reached, and at what point?
2. In general the discussion attempts to weigh the results with other literature in the field. I think this section could be improved with a more grounded focus on the ethnic minority group included in this study. I find no mention of specific cultural factors for black youth, the experiences of racism and discrimination, poverty, structural violence, experience of migration and stigma and aspects important to resilience such as social cohesion/social capital for black people in the UK. For example, this study found that young black adolescents had particular feelings of neglect and a lack of support from structures, this could be related to literature about barriers to accessing care – which can include shame, cultural identity, perceived communication skills of providers (one example from BMJ Open: Memon A, Taylor K, Mohebati LM, et al Perceived barriers to accessing mental health services among black and minority ethnic (BME) communities: a qualitative study in Southeast England BMJ Open 2016;6:e012337. doi: 10.1136/bmjopen-2016-012337). I think that the broad findings have been commented on, but more work is needed to really relate these findings to the specific ethnic minority group under study. Further, the authors may consider relating the particular resilience of this group of young black people to other literature.

Minor areas:

- 1 The background information is informative, relevant and useful for the reader to understand the context of this research. I would suggest the background could be further strengthened by focussing on evidence for individual ethnic groups, rather than using broad evidence for the entire ethnic minority population (it is very heterogenous). I note that the population that was interviewed in this particular study are from black communities, therefore providing contextual information about this particular group may be informative for the reader. For example, Proto, E., & Quintana-Domeque, C. (2021). COVID-19 and mental health deterioration by ethnicity and gender in the UK. PloS one, 16(1), e0244419. Found that particular ethnic groups are more vulnerable than others for mental health deterioration during the pandemic.
- 2 Similar to the above point, I think the background section also would benefit from discussion on the role of gender when considering ethnic minority vulnerability to mental distress.
- 3 Much research has shown that for certain ethnic minorities, stigma plays a significant role as a barrier to seeking help for mental health – this would warrant more discussion in the background section as this paper is interested in exploring the current needs of ethnic minority youth with respect to their mental wellbeing.
- 4 Page 8 Lines 50 to 60, there is a detailed hypothesis about likely negative impact of covid on mental health. This makes sense and is justifiable, however my understanding of IPA is that data is collected without a hypothesis in place, so as to analyse data without that influence. I would suggest that this hypothesis is not needed.

	5 Page 13 Lines 13 to 31, I note that the ethnicity of participants is provided, and I note one participants preferred not to say. From reading this paper I cannot identify how authors assigned ethnicity – was this self-declared by the participants? Were they given particular categories to pick from? Other:  1. Page 3 (abstract) Lines 7-12, I find this initial sentence a little confusing, the phrase 'how this is taking place' implies that this study is going to explore the mechanisms of physical and mental health vulnerability, which is not the case, consider re-wording. 2. Page 3 (abstract) Lines 24-25 'Ten 20-minute... interviews', in the body of the text the interviews are described as 15 minutes and the mean time was 14 mins, consider re-wording to ensure cohesive throughout paper 3. Page 9 Lines 10-11 says 'with the pandemic still largely unfolding' – I am not sure I would agree with this sentiment. I would be happy to review any revised manuscripts.
--	---

REVIEWER	Rastogi, Ritika Brigham and Women's Hospital
REVIEW RETURNED	27-Mar-2023

GENERAL COMMENTS	Thank you to the authors and editor for the opportunity to review this important qualitative study on Black/multiethnic adolescents' mental health and well-being during and after COVID-19-related lockdowns. The paper makes a key contribution to the study of minoritized adolescent mental health by adopting a qualitative approach and directly highlighting the words and lived experiences of youth. I recommend the manuscript for minor revisions with my comments below.  1. P7 L34 reads "Young people from ethnic minorities..." and I believe it should read "young people from ethnic minority groups." 2. P7 L44 the authors reference youth resilience (citations 26 and 27). Please expand on the findings of these papers. 3. P8 L20 the sentence reads, "...rarely examine the underlying nature of the relationships." To me, the word "relationships" evokes social relationships like friendships, family relationships. Perhaps a better word choice would be "associations"? 4. P8 L51 the authors cite "worrying patterns" in qualitative studies of young people's mental health. Please expand on the findings of prior research. 5. In superordinate Theme 1, the subtheme "symptoms of mental illness" does not seem particularly differentiated from the subtheme of "individual impact" to me. Are the symptoms of mental illness not a form of individual impact which the pandemic has had on young people's wellbeing? 6. In superordinate Theme 3, I wonder if the findings would be better reflected by the theme being named "Lack of support obtained..." or "limited support obtained..." rather than merely "Support obtained during the COVID-19 pandemic and current needs."
---

	7. The manuscript page numbers in the bottom right corner of each page are incorrect.
--	---

VERSION 1 – AUTHOR RESPONSE

Reviewer: 1

Dr. Ruth Webber, Bradford Institute for Health Research

Comments to the Author:

The impact of the Covid-19 pandemic on young people from black and mixed-ethnic groups' mental health: A qualitative study. REVIEW

This piece is competently written and makes a strong case for the need to hear about experiences of the impact of the pandemic on young people from minority ethnic backgrounds which are often excluded from the literature. The paper provides good context setting for the specific impact on minority ethnic groups. You are reflective about the strengths and limitations, and provide interesting discussion about implications for policy and practice. However, I feel that there are changes which need to be made in order for this to be publishable. My primary concerns relate to referencing.

- You write in paragraph 2 on page 8: 'Existing quantitative studies on the mental health of individuals from minority ethnic backgrounds are limited in that they very rarely examine the underlying nature of the relationships. To date, only one qualitative study has focused on the impact of Covid-19 on minority ethnic individuals to elucidate 'why' certain relationships exist.' Firstly, what relationships are you referring to? This is a confusing paragraph and loses the thread of what your paper is trying to do. Also just from a quick google scholar search I have found a range of studies that have in fact looked at this issue from a qualitative perspective (see below for just some examples). For a paper to be of a publishable standard it needs to demonstrate rigour of research of previous work in the field in order to highlight its contribution. As it stands, claiming only one qualitative study has looked at the link between ethnicity and the impact of covid-19 on mental health is untrue and suggests a lack of extensive reading.

- Response: Thank you for your suggestion. We have now made edits to this paragraph. We have rephrased the original sentence to clear up any ambiguity. We have also removed our statement that there is only one study to date examining ethnicity and mental health during Covid. What we meant to say was that very few qualitative studies recruited and focused exclusively on individuals from minority ethnic backgrounds to analyse and seek to understand their experiences in depth. After conducting another search, we have found the following article by Burgess et al. (Burgess, R. A., Kanu, N., Matthews, T., Mukotekwa, O., Smith-Gul, A., Yusuf, I., ... & Gul, M. (2022). Exploring experiences and impact of the COVID-19 pandemic on young racially minoritised people in the United Kingdom: A qualitative study. *Plos one*, 17(5), e0266504. <https://doi.org/10.1371/journal.pone.0266504>) that focuses on young people from ethnic minority backgrounds' mental health through focus groups. This was been published when the paper was initially written, but we have weaved this study into the rest of the paper to ensure the most updated references are included (see reference 30).

Thank you for suggesting the following papers.

o Shah, P., Hardy, J., Birken, M., Foye, U., Rowan Olive, R., Nyikavaranda, P., ... & NIHR Mental Health Policy Research Unit Covid coproduction research group. (2022). What has changed in the experiences of people with mental health problems during the COVID-19 pandemic: a coproduced, qualitative interview study. *Social Psychiatry and Psychiatric Epidemiology*, 57(6), 1291-1303.

- Response: Thank you for recommending this paper. After some discussion, we have decided not to include this study in our paper, as only a little over a third of all participants were from minority ethnic backgrounds, and all are adults, while we focus on young people. Although individuals from ethnic minority groups were actively recruited in the study and despite observing that these groups

have suffered from unequal impacts throughout the pandemic, the study does not delve deeper into the experiences of BAME participants specifically, nor does it consider specific issues they may face (which they explain is a limitation of their study). As it does not focus specifically on ethnic minorities as a group of vulnerable individuals and of young people, we have therefore decided not to include it in our paper.

o Gillard, S., Dare, C., Hardy, J., Nyikavaranda, P., Rowan Olive, R., Shah, P., ... & NIHR Mental Health Policy Research Unit Covid coproduction research group Katie Anderson Nick Barber Anjie Chhaphia Beverley Chipp Tamar Jeynes Jo Lomani Karen Machin Kati Turner. (2021). Experiences of living with mental health problems during the COVID-19 pandemic in the UK: a coproduced, participatory qualitative interview study. *Social psychiatry and psychiatric epidemiology*, 56, 1447-1457.

- Response: Thank you for recommending this paper. We had already included this study later on in the paragraph (now referenced as 31). The reason why we have not cited it as one of the very few studies focusing on ethnic minority and mental health qualitatively is because only 14% of the participants were Black/Black British. Additionally, all participants were adults. As it does not solely focus on individuals from ethnic minority backgrounds, but nonetheless contains relevant observations about their experiences with mental health, we have decided to keep it later on in the paragraph.

o <https://eur01.safelinks.protection.outlook.com/?url=https%3A%2F%2Fwww.mind.org.uk%2Fnews-campaigns%2Fnews%2Fexisting-inequalities-have-made-mental-health-of-bame-groups-worse-during-pandemic-says-mind%2F&data=05%7C01%7Ckeri.wong%40ucl.ac.uk%7Ca2ec734197f94166a2c708db2ee4b366%7C1faf88fea9984c5b93c9210a11d9a5c2%7C0%7C0%7C638155332014580852%7CUnknown%7CTWFpbGZsb3d8eyJWljiMC4wLjAwMDAiLCJQIjoiV2luMzliLCJBTiI6Iik1haWwiLCJXVCi6Mn0%3D%7C3000%7C%7C&sdata=J7U8aZobiHKXpmUUb6agmz1uVRwhBjvq5h60og%2FXYPY%3D&reserved=0>

- Response: Thank you for recommending this paper. This is the Mind charity survey we have cited throughout our paper (see reference 16). Although it provides interesting insights into the experiences of individuals from ethnic minority backgrounds with mental health across the pandemic, the sample is constituted at less than 5% of individuals from minority ethnic backgrounds. We have therefore decided not to include it here as we were looking for studies who focused exclusively on these marginalised groups.

o Pearcey, S., Burgess, L., Shum, A., Sajid, E., Sargent, M., Klampe, M. L., ... & Waite, P. (2023). How the COVID-19 pandemic affected young people's mental health and wellbeing in the UK: A qualitative study. *Journal of Adolescent Research*, 07435584231151902.

- Response: Thank you for suggesting this paper. Similarly, we have chosen not to include it in this section of the paper. We noted that the paper focuses on young people and mental health during the pandemic and provides interesting observations about people from ethnic minority groups being at the receiving end of abusive behaviour and blamed for the pandemic. However, these are not the main themes that we explored in our paper, and there was a lack of data focusing on participants from ethnic minority backgrounds.

o Scott, S. R., Rivera, K. M., Rushing, E., Manczak, E. M., Rozek, C. S., & Doom, J. R. (2021). "I hate this": A qualitative analysis of adolescents' self-reported challenges during the COVID-19 pandemic. *Journal of Adolescent Health*, 68(2), 262-269.

- Response: Thank you for suggesting this paper. After consideration, we have decided not to include it in this paragraph as it does not focus solely on young people from ethnic minorities, we

have found very interesting ethnic-specific data for black individuals that was relevant to other sections of our paper, that we have included throughout our paper (see reference 46).

- Furthermore, on P9 you say: 'The reality is that ethnic minority young people's mental health has not only been worsening since the start of the Covid-19 pandemic but is also poorly assisted [36, 37].' But if you claim only one study has looked at this, the claim you make here would not have enough evidence. Furthermore, neither reference 36 or 37 make any reference to minority ethnic young people's experiences.

- Response: Thank you for your suggestion. We did in fact notice that these studies that we had referenced here only mentioned the experiences of young people and not specifically ethnic minorities. To take on your suggestion, we have therefore corrected this and completed the references with another study depicting the experiences of young people from ethnic minority backgrounds.

- Again you talk about 'studies of the effect of the Covid-19 pandemic on ethnic minority groups' day-to-day life stressors', citing a qualitative piece of work by Mind (which I would explain who they are when you introduce them as knowledge that they are a UK-based mental health charity is currently assumed).

- Response: Thank you for picking this up. We have added more information on the Mind charity when we cite it in the third paragraph of our introduction. We hope our earlier clarification removes the confusion here.

- Finally, in the discussion you say that your findings: 'reinforces past research evidencing deteriorating mental health in young people from minority ethnic groups, both quantitatively [25, 28, 44] and qualitatively [26]', the very research you claim has not been done apart from one qualitative study. This is further evidence for the fact that your introductory discussion on existing work needs significant changes to be made, to acknowledge that there is qualitative work in this area, which as it stands it does not, yet goes on to seemingly reference work that has done this.

- Response: Thank you for your comment. We have rephrased this sentence to acknowledge the comparative scarcity of available qualitative literature delving into young people from minority ethnic backgrounds. Hopefully this clarifies any previous confusion.

- The sample size is very small (10) and the length of the interviews (20 minutes) renders the data collected very minimal. While qualitative research is by its nature small-scale and although you have clearly drawn patterns across the data, this seems particularly minimal, generating only 200 minutes worth of data. Again I can only speak from my background of anthropology, sociology and social policy, but this is a very limited amount of data to be able to draw conclusions from.

- Response: Thank you for your comment. We acknowledge that this is a very limited sample size and that hence the data obtained from it is also limited. However, we believe that the in-depth semi-structured interviews do yield valuable and insightful perspectives from minority ethnic young people that, as you can see from the literature, are not often represented. We agree that this is a starting point which can hopefully pave the way towards larger studies with wider and more representative samples, which we consider in our discussion of 'Implications for practice and research' (specifically in the second paragraph). We believe that the results and perspectives presented in this paper are relevant and paint a representative picture of the experiences of black and mixed ethnic young people in the UK that is worth exploring in more detail by future research, including other on-going work from a separate study from our lab.

- The structure of the paper, in particular the table's layout and use of themes and sub-themes almost identically mimics the qualitative paper that is the main focus by Mahmood et al. 2021. I am not from a health sciences background so it may be that this is standard practice, however you use the same quotes in 'connections' part of table 1 and 2 but where 1 is talking about during the lockdown, 2 is

talking about after the lockdown, so these quotes can't be used to say the same thing as they are talking about two separate time periods.

- Response: Thank you for spotting this. This was an error in the presentation of the tables and has been removed.

- You discuss different explanations for disproportionate impact of pandemic on minority ethnic populations, writing that: 'a biological explanation would suggest that ethnic minority groups have a higher risk of developing stress-related physiological responses and comorbidities [7, 8]. The explanations which follow this are much more robust in terms of addressing structural issues, while this first 'explanation' is very deterministic. Again I am not from a medical background but I'm not sure it has been claimed that minority ethnic groups are inherently more predisposed to poor mental health. I would delete this sentence and the following discussion is stronger, as this sentence suggests.

- Response: Thank you for your suggestion. We have rephrased this, as we agree that the ecological explanation is not only stronger, but supported by a strong body of evidence and is relevant to the rest of argument of this paper.

- You use the term 'co-production' in the conclusion but this is a very specific approach not used in this research and I would remove reference to this as this is the first and only mention of this and the research is not co-produced.

- Response: Thank you for picking this up. We agree that this term is not appropriate for the data included in this paper, as it was the wider intention of the co-production project, and we have decided to remove it.

Again, this paper is for the most part very well-written, but these issues above need to be addressed in order for it to be a publishable standard, and to ensure the continuity of a high standard of rigorous research.

- Response: Thank you for taking the time to share such detailed comments and suggestion to help us improve our manuscript even more. We appreciate your suggestions.

Reviewer: 2

Dr. Jordan Bamford, The University of Manchester Division of Psychology and Mental Health

Comments to the Author:

Dear Editor, thank you for the opportunity to peer review this paper.

The paper I have reviewed presents as a qualitative study which aims to explore the impact of the covid pandemic on ethnic minority youth mental health, and identify what support may best serve this population. This study included 10 participants recruited via convenience sampling in West London, a majority of the sample were black. Participants took part in semi-structured interviews. Interviews were analysed via an interpretative phenomenological framework. This study identified interesting insights into the impact of covid on mental health, in particular the individual impact on self-esteem, motivation, experience of anxiety, depression and eating disorder, exposure to trauma and isolation. Further this study identified changes in mental health since the end of lockdown, in particular aspects of personal growth and better coping is reported by the youth in this study. This study explored the support given during the pandemic, and despairingly shows that young people felt there was a lack of support and felt neglected. This study shows evidence for the negative impact of covid on young people's mental wellbeing, but also speaks to the resilience and coping strategies of young ethnic minority groups. This is an innovative paper as it explores the particular views of young black minorities which are often excluded in academic literature. Overall I think this paper is valuable and offers important insights. My academic background relates to determinants of ethnic minority mental health, so I have focussed my review on this area, and less on the qualitative methodology, although I

identified no significant concerns. I have a few suggestions relating to the background and discussion which I believe would strengthen this paper:

Major areas:

1. I noted in the methodology and results section there was no mention of saturation of themes with respect to this qualitative study. Can authors please indicate if saturation of themes was reached, and at what point?

- Response: Thank you for picking this up. We have now added this information in the methods, in the 'Data Analysis' section.

2. In general the discussion attempts to weigh the results with other literature in the field. I think this section could be improved with a more grounded focus on the ethnic minority group included in this study. I find no mention of specific cultural factors for black youth, the experiences of racism and discrimination, poverty, structural violence, experience of migration and stigma and aspects important to resilience such as social cohesion/social capital for black people in the UK. For example, this study found that young black adolescents had particular feelings of neglect and a lack of support from structures, this could be related to literature about barriers to accessing care – which can include shame, cultural identity, perceived communication skills of providers (one example from BMJ Open: Memon A, Taylor K, Mohebbati LM, et al Perceived barriers to accessing mental health services among black and minority ethnic (BME) communities: a qualitative study in Southeast England BMJ Open 2016;6:e012337. doi: 10.1136/bmjopen-2016-012337). I think that the broad findings have been commented on, but more work is needed to really relate these findings to the specific ethnic minority group under study. Further, the authors may consider relating the particular resilience of this group of young black people to other literature.

- Response: Thank you for your suggestion and references. We understand there is a lot of rich and interesting factors that can be covered when discussing this topic. In particular, we had included a discussion of barriers to access to care and mental health aid in this specific cultural group in the discussion, which we have developed further (4th paragraph of the discussion). We have also decided to add literature relating specifically to black and mixed ethnic young people to relate the findings to our sample, including the paper you have suggested and other literature (discussion paragraphs 3 and 4, and under 'implications for research', see citations 46, 49 and 53).

Minor areas:

1. The background information is informative, relevant and useful for the reader to understand the context of this research. I would suggest the background could be further strengthened by focussing on evidence for individual ethnic groups, rather than using broad evidence for the entire ethnic minority population (it is very heterogenous). I note that the population that was interviewed in this particular study are from black communities, therefore providing contextual information about this particular group may be informative for the reader. For example, Proto, E., & Quintana-Domeque, C. (2021). COVID-19 and mental health deterioration by ethnicity and gender in the UK. PloS one, 16(1), e0244419. Found that particular ethnic groups are more vulnerable than others for mental health deterioration during the pandemic.

- Response: Thank you for your suggestion. As stated above, we have included literature relating to black and mixed ethnic groups more specifically in the discussion. Additionally, we have added the study that you have suggested (reference 17) in the introduction as it was also very relevant to our literature review (paragraph 3).

2. Similar to the above point, I think the background section also would benefit from discussion on the role of gender when considering ethnic minority vulnerability to mental distress.

- Response: Thank you for your suggestion. We acknowledge that this is a limitation of our study as we do not discuss gendered patterns in our findings. We have recognised this in our

discussion (under limitations) and have further included a collection of studies which elucidate on interesting gender variations in ethnic minorities' mental health across the Covid-19 pandemic.

3. Much research has shown that for certain ethnic minorities, stigma plays a significant role as a barrier to seeking help for mental health – this would warrant more discussion in the background section as this paper is interested in exploring the current needs of ethnic minority youth with respect to their mental wellbeing.

- Response: Thank you for your suggestion. Taking your point into consideration, we have now included additional literature on this in our introduction (reference 36), as well as in the fourth paragraph of our discussion (reference 53). However, as stated later on in the paragraph, stigma was not something our participants mentioned as an obstacle to seeking mental health support.

4. Page 8 Lines 50 to 60, there is a detailed hypothesis about likely negative impact of covid on mental health. This makes sense and is justifiable, however my understanding of IPA is that data is collected without a hypothesis in place, so as to analyse data without that influence. I would suggest that this hypothesis is not needed.

- Response: Thank you for your suggestion. We have removed this from the introduction for clarity.

5. Page 13 Lines 13 to 31, I note that the ethnicity of participants is provided, and I note one participants preferred not to say. From reading this paper I cannot identify how authors assigned ethnicity – was this self-declared by the participants? Were they given particular categories to pick from?

- Response: Thank you for your suggestion. Participants self-declared the ethnic group they most identify with from a comprehensive list. We have added this information to the results.

6. Page 3 (abstract) Lines 7-12, I find this initial sentence a little confusing, the phrase 'how this is taking place' implies that this study is going to explore the mechanisms of physical and mental health vulnerability, which is not the case, consider re-wording.

- Response: Thank you for your suggestion. We have rephrased this in the abstract.

7. Page 3 (abstract) Lines 24-25 'Ten 20-minute... interviews', in the body of the text the interviews are described as 15 minutes and the mean time was 14 mins, consider re-wording to ensure cohesive throughout paper

- Response: Thank you for your suggestion. We have calculated that the average time for an interview was 14 minutes and 6 seconds. We have corrected our error, and for convenience purposes we have decided to uniformly refer to them as 15-minute interviews throughout the paper.

8. Page 9 Lines 10-11 says 'with the pandemic still largely unfolding' – I am not sure I would agree with this sentiment.

- Response: Thank you for your suggestion. We agree that this statement is no longer up to date, and have modified it.

I would be happy to review any revised manuscripts.

Reviewer: 3

Dr. Ritika Rastogi, Brigham and Women's Hospital

Comments to the Author:

Thank you to the authors and editor for the opportunity to review this important qualitative study on Black/multiethnic adolescents' mental health and well-being during and after COVID-19-related lockdowns. The paper makes a key contribution to the study of minoritized adolescent mental health

by adopting a qualitative approach and directly highlighting the words and lived experiences of youth. I recommend the manuscript for minor revisions with my comments below.

1. P7 L34 reads “Young people from ethnic minorities...” and I believe it should read “young people from ethnic minority groups.”

- Response: Thank you for your suggestion. We have amended this accordingly.

2. P7 L44 the authors reference youth resilience (citations 26 and 27). Please expand on the findings of these papers.

- Response: Thank you for your suggestion. We have added more information about the factors that seem to have fostered resilience in these studies.

3. P8 L20 the sentence reads, “...rarely examine the underlying nature of the relationships.” To me, the word “relationships” evokes social relationships like friendships, family relationships. Perhaps a better word choice would be “associations”?
changed

- Response: Thank you for your suggestion. We have taken this suggestion and made the appropriate amendments.

4. P8 L51 the authors cite “worrying patterns” in qualitative studies of young people’s mental health. Please expand on the findings of prior research.

- Response: Thank you for your suggestion. We have added information regarding the general themes identified in these studies.

5. In superordinate Theme 1, the subtheme “symptoms of mental illness” does not seem particularly differentiated from the subtheme of “individual impact” to me. Are the symptoms of mental illness not a form of individual impact which the pandemic has had on young people’s wellbeing?

- Response: Thank you for your comment. We agree that mental illness is a consequence of the pandemic that has impacted young people individually. However, we noted that, in our interviews, participants frequently referred to traumatic experiences and mental illness. We considered this relevant enough to expand upon and outline as a relevant and notable trend on their own, differentiating them from other more generalised experiences such as lower self-esteem or low motivation. We hope this clarifies any confusion.

6. In superordinate Theme 3, I wonder if the findings would be better reflected by the theme being named “Lack of support obtained...” or “limited support obtained...” rather than merely “Support obtained during the COVID-19 pandemic and current needs.”

- Response: Thank you for your suggestion. We have reviewed all the responses under this superordinate theme. We agree with you that most, if not all observations point to a lack of support rather than an evaluation of it. We have therefore decided to change the name of the theme.

7. The manuscript page numbers in the bottom right corner of each page are incorrect.

- Response: Thank you for your comment. This seems to be due to some formatting issue that occurred when submitting the manuscript. We have removed the numbers in the bottom right corner, and hope fixes the issue.

VERSION 2 – REVIEW

REVIEWER	Webber, Ruth Bradford Institute for Health Research
REVIEW RETURNED	17-Apr-2023

GENERAL COMMENTS	Thank you for the opportunity to review this paper. I am happy that the authors have responded in full to my comments and queries, and am happy for this paper to be published in its current form.
---

REVIEWER	Bamford, Jordan The University of Manchester Division of Psychology and Mental Health
-----------------	--

REVIEW RETURNED	20-Apr-2023
-------------

GENERAL COMMENTS	Dear Editor, Thank you for opportunity to re-review this paper. As previously mentioned, this paper presents a qualitative study exploring the impact of the covid pandemic on the mental health of ethnic minority youth, and explored what support may best serve this population. As previously described, I believe this paper is valuable and offers important insights into minority mental health which is often excluded in academic literature. On review of the authors response to my initial review, I am confident that I have no further recommendations at present. The authors have addressed each point I raised. In particular they have updated the methodology and referred to saturation of themes, they have bolstered the background section with up to date references. On reading the manuscript again, I am content in suggesting this paper is presently fit for publication.
--

REVIEWER	Rastogi, Ritika Brigham and Women's Hospital
-----------------	---

REVIEW RETURNED	20-Apr-2023
-------------

GENERAL COMMENTS	Thank you to the authors for addressing my comments. The additions which expand upon the literature cited have strengthened the paper and further contextualized not only the rationale for the study but also the implications of the findings.
--